# Cork Development: What Lies Within

**DOI:** 10.3390/plants11202671

**Published:** 2022-10-11

**Authors:** Rita Teresa Teixeira

**Affiliations:** BioISI—Biosystems & Integrative Sciences Institute, Faculty of Sciences, University of Lisboa, 1749-016 Lisboa, Portugal; rtteixeira@fc.ul.pt

**Keywords:** cork, exploitation, phellem, phellogen, stress, suberin

## Abstract

The cork layer present in all dicotyledonous plant species with radial growth is the result of the phellogen activity, a secondary meristem that produces phellem (cork) to the outside and phelloderm inwards. These three different tissues form the periderm, an efficient protective tissue working as a barrier against external factors such as environmental aggressions and pathogen attacks. The protective function offered by cork cells is mainly due to the abundance of suberin in their cell walls. Chemically, suberin is a complex aliphatic network of long chain fatty acids and alcohols with glycerol together with aromatic units. In most woody species growing in temperate climates, the first periderm is replaced by a new functional periderm upon a few years after being formed. One exception to this bark development can be found in cork oak (*Quercus suber*) which display a single periderm that grows continuously. *Quercus suber* stands by its thick cork layer development with continuous seasonal growth. Cork raw material has been exploited by man for centuries, especially in Portugal and Spain. Nowadays, its applications have widened vastly, from the most known product, stoppers, to purses or insulating materials used in so many industries, such as construction and car production. Research on how cork develops, and the effect environmental factors on cork oak trees is extremely important to maintain production of good-quality cork, and, by maintaining cork oak stands wealthy, we are preserving a very important ecosystem both by its biodiversity and its vital social and economic role in areas already showing a population declination.

## 1. Growth and Uses

The cork oak *Quercus suber* L. (Fagaceae) is an endemic tree of the Mediterranean basin growing mainly in Portugal, Spain, southern France and Italy, and northern Morocco. Cork oak stands, known as “montados” in Portugal and “dehesas” in Spain, are complex agro-silvo-pastoral systems managed by human labor to provide a habitat for a diverse wildlife and cultural ecosystems services. Management and exploitation of cork oak stands dates back to roman times and, throughout the centuries, Portugal alone has become the world leader of cork production (55% of world’s production) and cork transformation [1,2,3].

Cork possesses a set of properties, i.e., a low material density and an extremely low permeability to liquids and gases, it is biologically and chemically inert and mechanically elastic, and the conferring cork material provide high insulation and damping capacities [1]. Due to all properties of cork, it is used in a plethora of products such as sealants, agglomerates and composites that can be transformed into bottle stoppers, insulation and surfacing panels for construction and aeronautics, for pollutants absorbers, for clothing, fashion items and decorative furniture pieces [4,5]. Besides the specific uses for cork singled out in this review, other compounds can also be extracted from bark, which are later used in medicine, construction, chemistry, clothing, energy and biofuels [6]. Cork has also been found to work as biosorbents for heavy metals [7,8], oils [9] and aromatic hydrocarbons [10].

Cork can be extracted from the tree as a single plank because the phellogen (the meristematic layer producing cork outwardly) forms a continuous cylinder layer all around the trunk. Cork oak debarking for cork exploitation can only occur during a short time window from June till July (highly dependent on the environmental conditions), when phellogen cells are fully active and the new cells produced still display thin and fragile cell walls. As far as industrial requirements are concerned, for cork quality, the first cork produced by the tree and harvested, also known as virgin cork (Figure 1a), cannot be used for high end products because it is a hard-rough cork, very irregular in cell/material density and thickness. After the first cork, each debarking occurs every 9-12 years allowing the tree, the time necessary to regrow the tree’s outer periderm. This cork, used by industry, is called reproduction cork and displays enough thickness to allow punching natural cork stoppers (planks must be at least 27 mm thick) (Figure 1b). Regardless, all cork removed from the tree can be exploited commercially as raw material for final products as agglomerates. Only second reproduction cork and following harvests named “amadia” cork meets the quality standards necessary for stopper production demanded by wine companies [11] (Figure 1b). Stopper bottles are the product with the higher industrial revenue; however, the amount of available good-quality cork needed is becoming scarcer in the field, compromising cork manufacturing goals. For several years now, the transformation industry is developing new products made from lower quality cork and leftovers from stopper production, reducing the dependency on the wine sector [2,12]. Furthermore, in cork exploitation, there is a growing interest in understanding periderm development to improve plant resilience and as sinks for CO_2_ sequestration [13].

Albeit, the importance of cork products in countries like Portugal, which accounts for 2% of all exported goods, cork production has shown a reduction of 2.7% over the last decade [11,12] reflecting a cork oak decline associated with abiotic stresses phenomena such as prolonged drought events and ever more frequent and long-lasting heat waves [14]. These authors came to the conclusion that besides environment, tree genetics have a strong contribution to the cork chemical composition.

## 2. Biology and Adaptation

Several herbaceous species display a limited secondary growth, produced by interfascicular and fascicular meristems, therefore lacking phellogen and a periderm. Most herbaceous or woody plants display secondary growth of plant organs, which confers widening and is accomplished through the activation of two secondary meristems; the vascular cambium and the phellogen (cork cambium). The increase in diameter reflects the periclinal divisions on these two cambiums, producing derivatives towards both ends with further distinct specifications. The vascular cambium produces xylem inwardly and phloem outwardly resulting in stem and root thickening. This vascular growth forces the primary epidermis to break down with concomitant development of a complete periderm, a protective tissue of mature organs [15]. The periderm is composed of the phellogen, the phelloderm (produced by the phellogen inwards) and the phellem (cork) present outward of the phellogen [1] (Figure 2j,k). In most woody species growing in temperate climates, the first periderm is replaced by a new functional periderm a few years after being formed. This way, bark begins to accumulate on the outside comprising layers of dead periderms and remaining non-functional phloem tissue between them. This structure is called rhytidome [16]. One exception to this bark development can be found in *Quercus suber* which displays a single periderm that grows continuously. Such continuous growth of the phellem gives rise to cork rings that can be clearly distinguished because spring forming cells have thinner walls and a larger diameter compared to the cells formed later in the season [17] (Figure 1b).

The activity of the phellogen begins in April and continues to be active till the end of October [18]. During these months, growth is not uniform. In early season (April–July), cells are formed faster, thus display thinner walls and larger lumina diameter than cells growing later in the season (August–October). Such difference in cell anatomy permits distinguishing cork rings in a similar fashion to tree-rings (Figure 1b). Weather plays a role in cork-ring growth with a higher growth resisted in rainy years with moderate-low temperatures [19]. In fact, even outside of growing season (November–December), precipitation positively influences cork growth [20] as well as moderate temperatures during the summer [21]. However, it has been verified that high summer temperatures negatively influenced cork production [13]. The constrains that weather pressure exert on cork development is of great worry for cork producers and the cork industry. Costa et al. (2022) [22] reported that under water stress, smaller cork cells with reduced lumina and thicker cell-wall were produced resulting in denser cork rings. With climate changes hitting the Mediterranean area stronger than in many other zones of the globe, with the forecasted increased summer temperatures and longer, more extreme drought periods, it is very important deepen our understanding on the mechanisms governing cork development so that tree improvement programs can be put in place.

We ought to keep in mind that the periderm is first and foremost a protective tissue that works as the first line of defense of trees against abiotic and biotic stresses. The periderm confers resistance to pathogen invasion due to suberin deposits on cell walls and the presence of metabolites in cork tissues [23]. Since suberized cells are impervious, water vapor and the gases CO_2_ and O_2_ are regulated through lenticels. Lenticels are aerenchymatous structures composed mainly of active meristematic cells interspersing the phellem layer allowing gaseous exchange between the tree and the environment [24] (Figure 2k). Lenticels, unlike the neighboring suberized cells lack suberin but are essentially composed of lignin [1]. Lenticels arise from previous structures involved in gases exchange, the stomata, which are present in the epidermis and are genetically determined. For the cork industry, the presence of a high number of lenticels downgrades the cork quality, thus bringing its price down for producers (Figure 1c). Therefore, understanding the ontogeny of lenticels and the reason why some trees display a much higher number of these structures is of great importance to cork production.

## 3. Chemical Composition

The characteristic cork properties are a direct reflection of the cell wall’s chemical composition and structure. Cork’s chemical nature dictates its function as a protective layer of the internal tree’s tissues against environmental aggression, which is achieved by its main cell wall component, suberin. Other important cork cell wall components are lignin, the polysaccharide cellulose and the hemicelluloses and polar and non-polar extractives [1]. Cork cells present a wide chemical variation concerning suberin contents (23.1% to 54.2%) and lignin (17.1–36.4%) with the ratio suberin-to-lignin playing a pivotal role in physical cork properties such as compression [25,26].

Comparing the chemical composition of *Q. suber*’s cork, phloem and wood tissues, cork cells are the only ones with suberin on its cell walls (comprising 42.3%). Phloem on the other hand, shows a higher degree of lignification in comparison to wood and have less polysaccharides [27].

Suberin is made up of two domains: an aliphatic zone and an aromatic zone that includes ferulic acid [28]. One of the most recent accounts of suberin’s ferulic acid content was of 2.7% [29]. The polymeric aliphatic macromolecule is composed of two types of monomers: glycerol and long chain fatty acids and alcohols, whose hydroxyl and carboxylic groups are linked by ester bonds [30]. The monomer glycerol represents 40.8% of suberin and the long-chain monomers comprising mainly α-ω-diacides at 36.4% [31]. The fatty acids present chain lengths that can vary from C16 to C30, esterified to glycerol and cross-esterified [32]. At suberin’s macromolecular structural level, the polyester aliphatic structure is extensively linked to aromatic moieties [33] conferring the secondary cell wall, the appearance of lamellate structure as seen by transmission electron microscopy (TEM) [34]. These lamellae show alternate opaque and translucent contrasting structures, with translucent lamellae showing a regular thickness of ca. 30 Å and the opaque lamellae varying in thickness of ca. 70 to 100 Å. Between 30 and 60 lamellae have been counted in the suberized cell walls of cork of different species [34]. Recent work comparing the ultrastructural analysis of secondary cell walls in *Quercus suber*, *Quercus cerris*, *Calotropis procera* and *Solanum tuberosum* found that the lamellar structures with alternating dark and light bands were present in suberized cells of potato tuber periderm and *Calotropis* bark, whereas the cork cells of *Q. cerris* and *Q. suber* lacked defined lamellae [35]. Such findings led to the hypothesis that the chemical composition of suberin, which differs between species, may play a role in the cell wall topochemistry due to different spatial development of the suberin macromolecule [35].

The second most important cork cell wall component is lignin which confers strength to the cell. The remaining cell wall polysaccharides are cellulose and hemicellulose accounting for 20% of the cork’s cell wall structural components [36,37]. Lignin is present in most plants’ secondary tissues. It is a polymer comprising three different types of phenyl propane monomers, conferring an aromatic nature to lignin. The monomers ρ-coumaryl, coniferyl and sinapyl alcohols are linked through enzymatic phenoxy radical formation [1,30]. Lignin is also linked to ferulic acid by ester bonds accounting for 3% [29].

Non-structural components, soluble in different solvents, are also present in cork, such as lipophilic extractives that include fatty acids and alcohols, sterols and terpenes and phenolic compounds [6,38]. Ash is an inorganic material found in percentages ranging from 1% to 2% and is the end result upon total combustion. Amongst all ash components, calcium, phosphorous, sodium, potassium and magnesium are the minerals present in the highest concentrations [37,39]. In sum, in cork cells, the composition of the secondary cell wall depositions that takes place in the interior of the cell is of about 42% suberin, 22% of lignin, 19% polysaccharides and 16% of extractives [25]. The extractive portion is a myriad of low to medium molecular weight molecules and includes: n-alkanes, n-alkanols, waxes, triterpenes, fatty acids, glycerides, sterols, phenols and polyphenols that can be sorted in two main groups; aliphatic and phenolic [1]. Suberin, requires the presence of a lignin-like polymerized aromatic domain to adhere to the cell wall [40], whereas waxes are aliphatic compounds that do not covalently link to the cell wall [41]. Suberization is a fast process which can be visualized as soon as phellogen cells grow giving rise to a differentiated phellem (Figure 2j) [42]. Phellem development begins as soon as one-year stems grow (Figure 2a,d,g) and become thicker as the tree ages (Figure 2b,c,e,f,h,i). This can be detected on the roots and the aerial parts of the tree [43].

Polyphenolics are another group of extractives found in cork cells that are much less studied, with more heterogenous composition among phellems, which include simple phenols and/or polymeric phenols such as tannins. Out of the few studies on polyphenols in cork, Pinheiro et al. (2019) [44] were able to establish a comparison of phenolic compositions between cork samples of higher and lower quality findings that in cork of higher quality, aromatic phenylpropanoid components were incorporated into the cell wall in larger amounts than hydrolysable tannins. This work was carried out in parallel with a transcriptomic analysis using the same cork samples. Here, it was possible to see that during the development of cork of superior and inferior quality, and in the former, the shikimate pathway shifts towards the synthesis of cell wall-bound phenolics. In the latter, phellogenic cells invested more into the biosynthesis of soluble phenolics, especially hydrolysable tannins, displaying a higher reducing capacity (40% more than in cork of higher quality) [44,45].

## 4. Molecular Basis of Cork Development

From a transcriptomic point of view, at the beginning of the season, cell division and differentiation genes such as cyclins, are enriched, while later in the season, wall biogenesis and factors involved in secondary metabolites are elevated [15]. Genetics plays a decisive role in cork growth and its chemical composition [46,47]. High-throughput sequencing of cork oak phellogen samples isolated from thick and thin cork planks (examples in Figure 1b,c) collected on the same day, showed differences in the gene expression [45,48]. Trees producing a thinner cork layer and with higher number of lenticular channels (Figure 1c) exhibited a higher number of transcripts involved in DNA synthesis, RNA processing, proteolysis, and especially transcription factors associated with the abiotic stress response and stomatal/lenticular-associated genes. On the other hand, trees producing more uniform and thicker cork planks expressed genes encoding for heat-shock proteins were up-regulated [45]. When a transcriptomic analysis of phellogen cells isolated by laser microdissection was performed on samples collected from trees producing cork of good- and bad-quality, stress-related genes were enriched in samples from bad-quality cork which also showed an up-regulation of genes belonging to the flavonoid pathway [45]. These authors demonstrated that cork of lower quality displayed a higher content of free phenolics and, when analyzing the phenylpropanoid pathway, verified that in lower quality cork there was an up-regulation of genes involved in the biosynthesis of free phenolic compounds; while in good quality cork, the synthesis pathways for lignin and suberin was promoted [48]. Both these works showed that besides other environmental and genetic factors, specific gene expression patterns dictate which phenylpropanoid pathway branch is promoted (suberin synthesis or free phenolic production) resulting in differences in cork layer thickness. However, only part of the mechanism governing cork development can be explained purely by gene expression. For example, distinct epigenetic patterns and single-methylation events associated with climate variations influence cork quality [49]. Epigenetic gene regulation is done through the action of histones and DNA modifications which include methylation and acetylation. Basically, DNA and histone modification enzymes have the capacity to alter both euchromatin and heterochromatin conformations creating regions of condensed chromatin which physically prevents a given gene from being transcribed at a particular time point. Such regulation can be rapidly reversed every time a tree’s growth and/or environment conditions change, requiring differential gene expression. Epigenetic gene regulation processes are essential for plant development but have a special role when it comes to environmental adaptation, hence the abiotic/biotic stress responses. Recently, Silva et al. (2020) [50] identified in *Q. suber*, DNA methyltransferases (DNAMtases) and DNA demethylases (DDMEs) involved in histones methylation, demethylation, acetylation and deacetylation. In *Q. suber* phellogen cells, epigenetic-related genes were investigated to establish a relationship between epigenetic regulation and cork quality [51]. The research group found that all classes of DNA methyltransferases were present in *Q. suber* and that QsDRM2 was the methyltransferase most active in the phellogen. The specific DNA methylation patterns are pivotal for proper tissue establishment and differentiation [52,53]. Interestingly, expression of *QsDMAP1* (*DNA Methyltransferase 1-associated Protein 1*), involved in transcription repression and activation and in genomic instability [54], was higher in cork of low quality [51], suggesting that cork inclusion could be a mechanism for the tree to cope with the environment. During heat-stress experiments, Correia et al. (2013) [55] showed that DNA methylation and histone H3 acetylation acted contrary to each other, whereas DNA methylation increased in *Q. suber* plants as temperatures rose from 35 °C to 55 °C under a controlled environment. On the other hand, *acetylated histone H3* (*AcH3*) levels decreased during the same temperature range experiment.

Besides adaptation to stress conditions, epigenetic gene regulation also plays an important role on well-programmed seasonally regulated growth conditions, such as dormancy. Trees growing in temperate and boreal forests have developed an elaborate mechanism to survive the cold temperatures during winter and the capacity to resume growth once spring conditions arrive [56]. As the photoperiod diminish, a set of gene responsible for dormancy regulation mechanisms are initiated [57] which are highly dependent on epigenetic gene silencing [58]. The epigenetic regulation of *DORMANCY-ASSOCIATED MADS-box* (*DAM*)-related genes and *FLOWERING LOCUS C* gene are two examples of genome-wide epigenetic modifications associated with dormancy events [58,59].

Upon transcriptomic analysis of cork oak phellogen samples collected in April (cork active growing season) it was possible to observe that meristem-associated genes, such as *WUSCHEL* (*WUS*), *SHOOT MERISTEMLESS* (*STM*), *APETALA2* (*AP2*), *PHABULOSA* (*PHB*), *AINTEGUMENTA* (*ANT*) and *AINTEGUMENTA-LIKE6* (*AIL6*) were up-regulated. Later on, during the active growth season (June till July) cell wall genes linked to lignin, suberin and secondary cell wall deposition were highly expressed [15]. The transition of gene expression from meristem identity/maintenance genes toward genes encoding elements necessary for secondary wall establishment reflects the developmental nature of cork layer from the phellogen. More recent works have contributed to elucidate gene activation involved in the phellogen, e.g., chromatin-remodeling genes and other genes already known to be involved in meristem regulation, such as *FLOWERING LOCUS C* (*FLC*) or stem-cell maintenance-related transcription factors [47,60]. The transcriptome of potato phellogen showed that genes associated with cell niche and radial patterning, such as the transcription factor *SHORT-ROOT 2B* (*TPR-2*), *NO APICAL MWEISTEM* (*NAM*), *HD2B*, *PHAVOLUTA-Like HD-ZIPIII*, and *PHLOEM INTERCALATED WITH XYLEM* (*PXY*) were present and highly expressed [61]. This study on potato phellogen showed an up-regulation of genes related to ribosomal proteins as well as stress-associated genes, including heat-shock proteins, a feature also observed in cork oak phellogen [45].

When it comes to cork phellogen regulation, phytohormones, auxins and brassinosteroids, seem to play an important role. It was shown that increased auxin levels proceeded ethylene production which, in turn, worked as a principal activator of genes involved in the phellogen’s initial development [62]. A putative *PIN3* orthologue, an auxin transporter, and *DWF1*, encoding an enzyme involved in the synthesis of brassinosteroid, are up regulated in cork phellogenic cells [47]. *ABA responsive element-binding protein/ABA binding factor* (*AREB/ABF*) and *MYC/MYB* transcription factors together with ABA-related genes displayed a higher number of elements up regulated in trees with a more pronounced cork layer development [45]. Additionally, in these cork samples, genes encoding ethylene-responsive transcription factors; *EIN3* (*Ethylene insensitive 3*) and *ETR1/ETR2* and *EIN4* (ethylene receptors) were also highly expressed [48]. These genes are membrane-associated receptors with the capacity to bind ethylene further triggering the downstream ethylene response cascade [63]. Lopes et al. (2019) [64] also detected an upregulation of the *AP2/ERF* ethylene responsive transcription factor in cork phellogen samples. Besides ethylene response-associated genes important in developmental mechanisms [65], jasmonate-associated genes, important in plant defense [66], were detected to be up regulated in cork oak [64]. Other jasmonate signaling-associated genes like jasmonate-ZIM domain-containing proteins and the jasmonate signaling mediator *MYC* transcription factor, were up regulated in thick cork samples suggesting the participation of jasmonates in stress responses during proper cork development [48]. Transcription factors are also regulated by abiotic stresses in cork oak phellogen. Consistently, the Arabidopsis ortholog *MYB84*, *QsMybB1* is down regulated in response to drought and heat [67,68] (Almeida et al., 2013a,b) and was shown to accumulate in poplar phellem [60].

Cork oak phellogen has the capacity to produce thick suber layers at a rate of 1–3 mm/year with pronounced suberin secondary deposition on the cell wall. One of the genes responsible for the suberization process is *FHT* (*fatty ω-hydroxyacid/fatty alcohol hydroxycinnamayl transferase*) which is specifically expressed in cells undergoing suberization regulated by ABA and salicylic acid (SA) [69]. Its activation is detected in phellogen cells right before suberin deposition is initiated [69]. *FHT* is also involved in wax biosynthesis, essential for periderm integrity [70]. The number of differently expressed genes involved in the phenylpropanoid pathway during cork development is elevated. Especially in suberin synthesis and assembling, several transcripts associated with fatty acid elongation are consistently found to be expressed, i.e., *3-ketoacyl-CoA synthesis* (*KCSs*); *long-chain acyl-CoA synthase* (*LACSs*), *fatty acyl-CoA reductases* (*FAR*) and *glycerol-3-phosphate acyltransferases* (*GTPAs*) [45,64]. Besides the detection of genes related to wax biosynthesis in cork oak phellogen, the suberin biosynthesis genes *GPAT5*, *CYP86A1*, *CYP86B1* and *AtHHT/ASFT*, *CYP86A33* and *FHT* were also found to be up regulated compared to holm oak *Quercus ilex* L. (not producing a cork layer but a rhytidome) [47]. All genetic analyses and tissues producing suberin have shown expression of cytochrome P450-family members involved in the catalysis of ω-hydroxylation of fatty acids necessary for suberin monomer production [71]. In fact, the cytochrome P450 transcripts *CYP86A1* and *CYP86B1*, *CYP72A*, *CYP81E*, *CYP82A* and *CYP87A* were identified in cork tissues [45,48,72,73]. Recently, Lopes et al., (2019) identified *CYP82D*, *CYP76A* and *CYP705A* in phellogen samples. When Arabidopsis root phellem was studied for suberin synthesis, it was possible to detect the expression of suberin synthesis components; *CYP86A1*, *CYP86B1*, *FAR1/4/5*, *GTPAT5/7*, *FACT*, and *ASFT* [74] indicating a conservative mechanism for suberin biosynthesis.

Albeit suberin synthesis confers singularity to suberin tissues, the most transcripts enriched category in cork oak phellogenic cell was “phenylpropanoid metabolic processes”, belonging to the phenylpropanoid pathway [47]. Genes encoding key enzymes of this pathway such as *phenylalanine ammonia-lyase* (*PAL*), *cinnamate-4-hydroxylase* (*C4H*) *and 4-coumarate-CoA ligase* (*4CL*) were up-regulated in cork samples. In potato leaves that have been mechanically injured, *PAL* expression peaked 2 h after wounding then returned to a basal level 6 h later [75]. PAL seems to require the action of ABA since low levels of the phytohormone leads to a reduction of suberin aromatic components by affecting PAL activation [76]. Further down the phenylpropanoid pathway acting in the early steps of the lignin branch, the enzyme hydroxycinnamoyltransferase (HCT) whose corresponding gene, when down-regulated, leads to a strong decrease of suberin-linked ferulic acid levels [77]. Proanthocyanidins (PA), the end product of the flavonoid pathway, a phenylpropanoid pathway branch [78], accumulates in the vacuoles of cells of the traumatic periderms and young cork cells, and associate with the cell walls of suberized empty cells [79]. Anatomy studies have already shown us that the first cork cells formed can be distinguished by their electrodense fillings and brown inclusion of tannins [18], a feature also observed in cork oak’s one-year stems [42] (Figure 2j). Tannins are part of proanthocyanidin (PA) secondary metabolites [80]. Both PA and hydrolysable tannins accumulate inside the vacuoles [80] and have been found in *Q. suber* [44,81] contributing to the plant’s defense mechanism [81]. The biosynthesis of PA components is achieved by the action of three enzymes: leucoanthocyanidin reductase (LAR), anthocyanidin reductase (BAN/ANR), two NADPH-dependent enzymes [82,83] and leucoanthocyanidin dioxygenase/anthocyanidin synthase (LDOX/ANS) and found to be up-regulated on cork samples with higher amounts of free phenolics [48].

Suberin, lignin and polyphenol production are amongst the most enriched categories during cork tissue development, but autophagy and programmed cell death (PCD) are also categories with high number of up-regulated genes [79]. Besides the high suberin content of periderm cork cells, another characteristic of these cells is the fact they are dead by the time functional maturity is reached, implying they undergo PCD [84]. In general, plant PCD is carried out by caspase-like enzymes and proteasomes degrading ubiquitination proteins [85]. Particularly during periderm development, PCD has been regarded as a crucial event for cell maturation and a set of PCD marker genes [86] are exclusively expressed in the endodermis and dying phellem cells in Arabidopsis during periderm formation [87]. Focusing on the periderm, PCD begins simultaneously as soon as fast cell wall suberization begins and phellem cells differentiate [18,88]. Programmed cell death occurring in cork oak has a developmental (dPCD) nature rather than being an effect of environmental factors [79].

Regardless lenticels support the exchange of vital gases, such as CO_2_, O_2_ and water vapor between the inside and outside [24] of the stem, the presence of lenticel channels affects the quality and price of natural cork stoppers since cork planks quality, is defined according to the external surface porosity promoted by these lenticels (Figure 1b,c). A higher number of lenticels is indicative of a lower quality of the cork (Figure 1c) [5]. Lenticular phellogen differentiation leads to the development of unsuberized cells, which are mainly composed of lignin [1]. Their development has a genetic background comprising at least three basic helix–loop–helix (bHLH) transcription factors forming a single clade [89,90], i.e., *SPEECHLESS* (*SPCH*), *MUTE*, and *FAMA*, identified in Arabidopsis and essential for stomatal formation [91,92]. When a comparative transcriptomic analysis was carried out on phellogen from good- and low-quality cork samples, it was possible to observe an up regulation of two orthologues included in the same bHLH clade in low-quality cork, samples characterized by the present of a much higher number of lenticular channels [45] (Figure 1c).

More and more attention has been directed to cell proteomes but when it comes to cork oak suberin development, the amount of work reported is practically non-existent. One of the first reports was conducted by Ricardo et al. (2011) where it was found that the most significant proteins associated with cork development were “carbohydrate metabolism” (28%), “defense” (22%), “protein folding, stability and degradation” (19%), “regulation/signaling” (11%), “secondary metabolism” (9%), “energy metabolism” (6%), and “membrane transport” (2%). Recently, the work performed on phellogenic cells isolated from fresh debarked planks showed high amounts of oxidoreductases (26%) and metal ion binding proteins (16%) [44]. Examples include, dehydrogenases and enolases (involved in glycolysis and respiration) and signaling-related proteins, such as annexins and other Ca^2+^-binding proteins [44]. In accordance with what has been observed in numerous transcriptomic analyses during cork development, the proteomic study carried out by Ricardo et al. (2011) [93] indicated a high involvement of defense proteins (thioredoxin-dependent peroxidase, glutathione-S-transferase, SGT1 protein, cystatin, and chitinases) in phellem differentiation, indicating that protection of tree’s inner tissues is the main function of the cork layer.

## 5. Conclusions

Knowledge on periderm development has broadened in these last years due to its protective role especially when coping with a climate emergency. It has now been regarded as not only as a physical shield of the plant’s internal tissues, but also as a tissue that accumulates secondary metabolites which can be used in industry for various proposes. This review focused on *Quercus suber* for displaying the formation of a continuous subereous layer that has been exploited by man for thousands of years. Nowadays, everyone is familiar with cork stoppers, the cork product that still carries out the most revenue for the transformation industry. Cork oak allows the exploitation of its raw cork layer while keeping the tree alive permitting silviculture practices on cork oak stands to deliver a win-win situation at all levels. Ecologically because cork oak stands harbor a rich ecosystem in terms of biodiversity and also works as powerful CO_2_ sequestering sinkers. Additionally, it’s an important social player since cork exploitation enables economically profitable activities in regions where human population is declining fast. To some extent, keeping cork oak trees healthy works as a buffer against putative soil and human desertification. After so many centuries protecting and exploiting cork oak trees empirically, it is time scientific knowledge to step in and contribute toward an even better sustainable silviculture practices using molecular markers for improved clone selection which can growth with a better successful rate under stressful conditions and continue to provide cork of good quality which is demanded by industry. 

## Figures and Tables

**Figure 1 plants-11-02671-f001:**
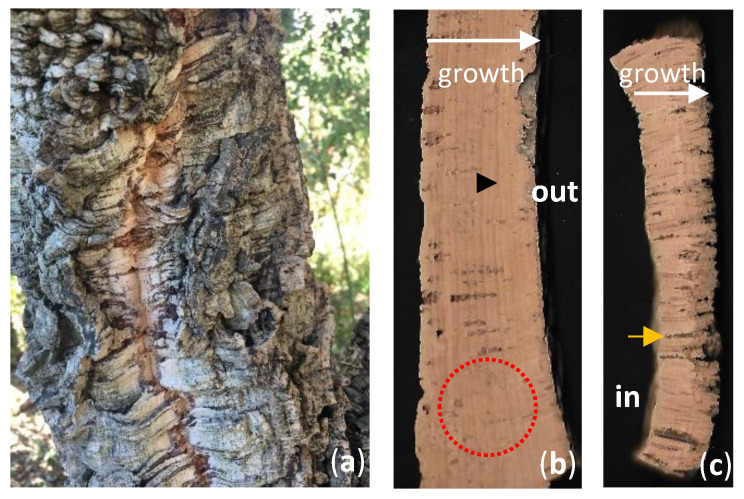
The cork layer of *Quercus suber*. (**a**) Virgin cork. This cork oak has never been debarked is still displaying its first cork layer. (**b**,**c**) Different quality types of *amadia* cork. (**b**) Cork plank thick enough and with very few discontinuities allowing stopper punching (red dots circle). It is possible to distinguish the growth rings (black arrow). (**c**) Example of a reduced-quality cork plank, thin in thickness and intersected by numerous lenticular channels (black arrowhead). White arrows indicate the growth direction from the surface in contact with the trunk and where the phellogen is (also known as cork belly) towards the surface in contact with the environment. **In**—cork plant inside; **out**—outermost cork plank surface.

**Figure 2 plants-11-02671-f002:**
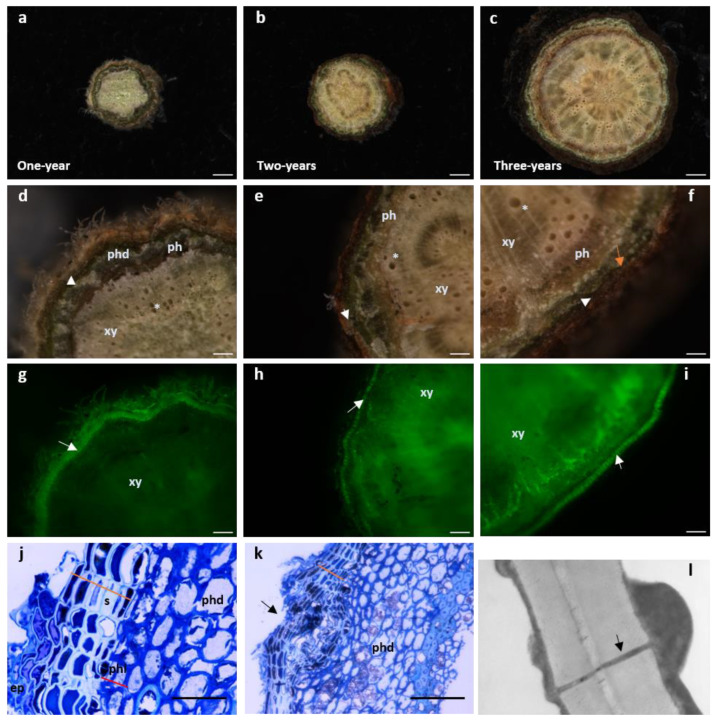
Cork develops as early as one-year stems in *Quercus suber*. (**a**–**i**) Cross-sectional stereomicroscopic images of one-, two- and three-year-old stems. (Scale bars: (**a**,**b**) 100 µm, c 500 µm). (**d**–**f**) Details of the xylem with empty tracheids (*****). Moving outwardly, one finds the phloem (dark layer) surrounded by the periderm comprising the phelloderm (orange arrow -**phd**), a thin layer of phellogenic cells (white arrowhead) and several rows of suber cells. The most external layer is the remaining epidermis. (Scale bars: (**d**,**e**) 100 µm, f 500 µm). (**g**–**i**) Cork cells exhibit autofluorescence when excited with UV light [18]. The stems observed in d-f were excited with UV using a stereoscope filter for “Lumar01” (BP 365/12; LP397) after which, the suberin layer is clearly distinguishable (white arrow). In older stems, as xylem cells mature, autofluorescence can be observed since lignin also possesses some autofluorescence properties under UV light. (**j**,**k**) Light microscopy of one-year stem cross sections stained with toluidine blue. (Scale bars: (**g**,**h**) 100 µm, i 500 µm). (**j**) Some remaining epidermis (ep) is still present and right below, six to seven cell cork layers with cells displaying a hyaline cell wall appearance and filled with electrodense material (phenolic compounds) (orange line) can be observed. Inwardly and adjacent to suberin cells, is the phellogen composed of only one to two cells (red line). Cells are filled with cytoplasm and cell walls do not present a secondary thickening. Right below is the phelloderm with the characteristic round-shaped parenchymatous cells. (Scale bar: 10 µm). (**k**) Cross section showing a lenticel forming in a one-year stem (black arrow). The disorganized division of the meristematic cells inside the lenticel structure is starting to push the suberin layer outwards leading to a complete rupture of this layer to form an aperture that allows gas exchanges. (Scale bar: 40 µm). (**l**) Transmission electron microscopy image of *amadia* cork cell wall clearly showing a plasmodesmata (black arrow) crossing both suberized cell walls. It is possible to observe cytoplasmic deposition in the inner part of the cell, with thickening appearing on both sides of the plasmodesmata channel (amp. 20,000×). **ep**—epidermis; **ph**—phloem; **phd**—phelloderm; **phl**—phellogen; **s**—suberin; **xy**—xylem.

## Data Availability

Not applicable.

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
