# Peer review of "Cork Development: What Lies Within"

_plants, 2022, doi:10.3390/plants11202671_

Round 1

Author Response

Plants

Manuscript 1888421

Title: Cork development: what lies within

Author: R T Teixeira

Reply to reviewer 1

I would like to deeply thank for taking the time and careful reading of the manuscript and for all the comments which increased greatly the quality of the text.

I introduced all the suggestion in the text even though some of them might be masked for the extra text added and comments from other reviewers.

I introduced a long paragraph about epigenetics, a subject very important that was lacking in this manuscript. I also wrote a final conclusion because in fact, was missing at the end of all the information.

Author Response

Plants

Manuscript 1888421

Title: Cork development: what lies within

Author: R T Teixeira

Reply to reviewer 2

I would like to deeply thank for taking the time and careful reading of the manuscript and for all the comments which increased greatly the quality of the text. I’m particularly appreciative for the special care put into all the details such as the figure legend. It was a precious input. 

I introduced all the suggestion in the text even though some of them might be masked for the extra text added and comments from other reviewers.

The next specific answers try to explain some aspects I was not able to fix either because it was not managed within the scope of this manuscript or because technically, there was some constrains preventing me from getting the optimal image. Despite my explanations, the reviewer comments are extremely valuable for any researcher.

The description of genetic backgrounds could still be perfected by including a schematic

drawing illustrating the described processes, their interrelationships, and corresponding

plant localities

R: Creating a scheme is very appealing which I have tried to do but at this point, the amount of information already available yet still full of black boxes turned out to be a very complicated task. I felt I was introducing errors and misleading the readers. I will work on this with time for future submission, for sure. Regardless, I really appreciated the suggestion which have sown the will to get such a scheme.

 Do all the described genetic/metabolic processes match the „native“ development of cork;

i.e., the cork was not biotically and/or abiotically impacted/altered or harvested? Thus, the

question arises of how the genetic/metabolic processes would run after cork harvest and

other alterations of cork?

R: working with cork development is challenging because suber cork cells are dead and for that reason, phellogen is the tissue normally used for genetic/molecular analysis. Since cork is part of a protective tissue, it is susceptible to distinct gene regulations accordingly to the biotic/abiotic stress it is submitted to. This aspect is mentioned several times throughout the manuscript but it not my intuit to dissect this problematic but rather provide referents that could be used by the reader. Nevertheless, it is a debate permanently present in my investigation/line of thought.

Knowing about the genetic backgrounds, does the cork quality and development differ

between different oak populations/genotypes (e.g., at various locations, different genetic

properties) of the same oak species?

R: yes, it does and it is well known by the cork producers. Practically all cork oak stands are of natural origin meaning that trees at a given area ought to have the same genetic background. When comparing cork quality, we always collect samples from different stands/areas of the country. These procedures are always well explained in the research literature used to write this manuscript that can be specifically exploited depending on the question that needs to answered by the reader.

Subfigures 2 a-f, k seem to be out of focus – could they be improved by contrast and/or

sharpening, e.g., via an bioimage software?

R: you are absolutely right but it was the compromised found between the struggle of getting small enough pieces of stem without destroying them with the razor blade. I wanted to use exactly the same settings in all the images to reduce any noise amongst the three samples so I focused on the periderm bring out of focus the xylem whenever I couldn’t prevent it. 

Reviewer 3 Report

This paper reports on the Cork development: what lies within”. The article is interesting, but too short. It needs to be extended. There is no clearly defined goal and summary. Corresponding parts of the article template were not preserved. The whole thing needs to be expanded and improved.

Taking into account all comments the manuscript may be published in Plants after major revision.

Author Response

(The authors gave the same response as above.)

Round 2

Reviewer 3 Report

Manuscript can be published in prezenter form.